# Vein of Galen Aneurysmal Malformation: A Case Report

**DOI:** 10.3390/healthcare12070716

**Published:** 2024-03-25

**Authors:** Naomi E. Clarke, Jatinder Shekhawat, Himanshu Popat, David J. E. Lord, Mohamed E. Abdel-Latif

**Affiliations:** 1National Centre for Epidemiology and Population Health, Australian National University, Acton, ACT 2601, Australia; 2Kirby Institute, University of New South Wales, Sydney, NSW 2052, Australia; 3Department of Radiology, Canberra Hospital, Garran, ACT 2605, Australia; 4Garran Medical Imaging, Garran, ACT 2605, Australia; 5The Children’s Hospital at Westmead, Westmead, NSW 2145, Australia; 6Sydney Medical School, University of Sydney, Westmead, NSW 2050, Australia; 7Department of Neonatology, Centenary Hospital for Women and Children, Canberra Hospital, Garran, ACT 2605, Australia; 8Discipline of Neonatology, School of Medicine and Psychology, College of Health and Medicine, Australian National University, Acton, ACT 2601, Australia; 9Department of Public Health, La Trobe University, Bundoora, VIC 3083, Australia

**Keywords:** vein of Galen aneurysmal malformation, neonatal cardiac failure

## Abstract

Vein of Galen aneurysmal malformation is a relatively rare disease in which failure of the median prosencephalic vein of Markowski to involute early in gestation leads to a grossly dilated deep cerebral vein with multiple arterial feeders, causing a large arteriovenous shunt which leads to high-output cardiac failure. We describe a case of a term neonate who presented to a tertiary neonatal centre on day one of life with history, symptoms, and signs consistent with perinatal asphyxia; however, in the context of worsening multi-organ dysfunction and cardiomegaly, the infant was found to have a severe vein of Galen aneurysmal dilatation leading to high-output cardiac failure. The patient was transferred to a tertiary paediatric hospital and underwent a total of four coiling procedures to embolise the multiple feeder arteries supplying the aneurysmal malformation. This case highlights the difficulties in diagnosing this relatively uncommon condition, particularly in the context of a possible perinatal insult.

## 1. Introduction

Vein of Galen aneurysmal malformation (VGAM) is a relatively rare disease, occurring in less than 1 in 25,000 deliveries and representing less than 1% of all intracranial arteriovenous malformations [1]. It is caused by a failure of the median prosencephalic vein of Markowski to involute early in gestation. This embryonic vein usually involutes by the 11th week of gestation, and its remnants join with internal cerebral veins to form the great cerebral vein (also known as the vein of Galen). The persistence of the median prosencephalic vein of Markowski leads to a grossly dilated vein with multiple arterial feeders. This causes a large arteriovenous shunt, with up to 70% of cardiac output directed to cerebral circulation [2]. The aetiology of VGAM is unknown; however, anomalies of both the RASA1 gene and the endoglin gene have been reported as potential causative factors [3,4].

We describe a case of a term neonate who presented to a tertiary neonatal centre with VGAM following CARE (case report) guidelines [5].

## 2. Case Report

A term male infant presented for review at a tertiary neonatal centre at 18 h of age. He had been delivered cephalically at home to a 29-year-old primigravida mother following an uncomplicated pregnancy. The maternal blood group was O negative, with anti-D given appropriately. Ultrasounds at 12 and 20 weeks were normal. Group B streptococcus testing was not performed. Delivery followed spontaneous onset of labour at 40 weeks, with clear liquor and no prolonged rupture of membranes. A midwife attended the planned delivery at home. Apgar scores were 4, 5, and 8 at 1, 5, and 10 min of age, respectively. Intermittent positive pressure ventilation was given for three minutes after delivery. Birth weight was 3760 g (57th percentile), length was 54 cm (88th percentile), and head circumference was 38.2 cm (98th percentile).

On examination at 18 h of age, the infant was mottled and had subcostal and intercostal recession. Pre-ductal oxygen saturation was 88%, with a respiratory rate of 70 breaths per minute. He was afebrile and normotensive, with a normal cardiovascular examination. Breath sounds were audible bilaterally. The abdominal examination was unremarkable. The tone was normal, and the infant was not encephalopathic.

Full blood count, electrolytes, and inflammatory markers were normal on admission. Capillary blood gas showed metabolic acidosis (pH 7.24, pCO_2_ 40, HCO_3_^−^ 16.1 mmol/L, base excess −10.3, lactate 3.4 mmol/L). Chest X-ray showed cardiomegaly with no focal collapse or consolidation (Figure 1). The echocardiogram was normal, apart from a 2.9 mm patent ductus arteriosus with bidirectional flow and mild suprasystemic pulmonary hypertension.

Differential diagnoses were sepsis, perinatal asphyxia, respiratory distress syndrome, and transient tachypnoea of the newborn. Initial management included continuous positive airway pressure (CPAP), oxygen to maintain pre-ductal SaO_2_ ≥ 95%, antibiotics (benzylpenicillin and gentamicin), and maintenance of intravenous fluids.

The patient’s oxygen requirement resolved within 48 h, but he had ongoing work of breathing requiring CPAP and persistent metabolic acidosis. He developed coagulopathy (INR 3.3, APTT 66 s, PT 37 s), renal impairment (urea 6.8 mmol/L, creatinine 99 µmol/L), and hyponatraemia (Na 126 mmol/L), and became oliguric and oedematous. Blood cultures were negative, but the umbilical swab grew Group B streptococcus. Additional management included intravenous vitamin K, fresh frozen plasma, and a sideline of concentrated sodium chloride. The working diagnosis was perinatal asphyxia given the acidosis and multi-organ dysfunction.

On day 5 of life, the patient deteriorated with severe respiratory distress, poor perfusion, minimal responsiveness, oliguria, and hyponatraemia (Na 117 mmol/L). Examination revealed worsening generalized oedema, ascites, and hepatomegaly. This was attributed to congestive heart failure. Chest X-ray showed worsening cardiomegaly. The patient was managed with intubation and ventilation, fluid restriction, frusemide, bicarbonate correction, and the addition of cefotaxime.

Following the stabilisation of the patient, a cranial ultrasound was performed, which showed a large anechoic structure mainly to the left of the midline posteriorly, with marked internal vascularity and prominent flow in keeping with the vein of Galen aneurysmal malformation (VGAM). The dilated median prosencephalic vein measured 36 × 26 × 29 mm (Figure 2). Clinical examination at this stage revealed cranial bruit, which was missed on admission.

The patient was transferred to a tertiary paediatric hospital for paediatric neurosurgery and intervention radiology assessment, as this service was not available at our hospital. Following the transfer, a brain MRI was performed (Figure 3), and the patient underwent a coiling procedure to embolise the arterial feeders of the VGAM. In total, four coiling procedures were performed over a period of six weeks. Following the second of these, the patient was found to have hydrocephalus and underwent insertion of an external ventricular drain, which was removed after several weeks. Before discharge, an MRI of the brain demonstrated a reduction in the size of the VGAM and arterial collaterals, but also showed ongoing hydrocephalus and small bilateral infarcts in the corona radiata.

The patient was discharged at seven weeks of age. The discharge head circumference was 40.5 cm (63rd percentile). At three months of age, the head circumference was 43.7 cm (90th percentile), and the weight was 5.7 kg (20th percentile). Development was appropriate for the patient’s age, although gross and fine motor scores were in the lower average range. MRI showed a mild progressive increase in the size of the VGAM, along with multiple collaterals and mild progressive dilatation of the ventricles. The patient may require multiple further embolisation procedures, and his prognosis remains uncertain.

## 3. Discussion

Vein of Galen malformation (VGAM) is a rare type of abnormality that affects the blood vessels within the brain. It develops as a direct arteriovenous fistula between the choroidal or quadrigeminal arteries and an overlying single median venous sac [6]. The venous sac most probably represents the persistence of the embryonic median prosencephalic vein of Markowski, not the vein of Galen, per se [6]. These malformations are rare, representing less than 1% of all intracranial arteriovenous malformations [7].

Antenatally, the diagnosis of VGM is well documented via colour Doppler ultrasonography [7,8] and foetal magnetic resonance (MR) [9].

The most common presentation of VGAM is neonatal high-output cardiac failure, as observed in our patient [10]. Less severe forms of the disease may present later in infancy and childhood, typically with hydrocephalus and macrocephaly, developmental delay, headaches, and/or seizures [11]. In neonates presenting with high-output cardiac failure, persistent pulmonary hypertension is common, and patients often develop multi-organ dysfunction, including myocardial ischaemia, which may worsen cardiac dysfunction. Persistent venous congestion may lead to cerebral atrophy and irreversible brain damage [11].

Most heart failure in the neonatal period is cardiac in origin, most commonly caused by congenital cardiac lesions, and less commonly arrhythmias, myocardial ischaemia, myocarditis, or cardiomyopathy. However, high-output cardiac failure represents another important cause. This is usually due to systemic arteriovenous shunts, including liver lesions (such as rapid involuting congenital haemangioma) and cerebral VGAM.

Foetuses prenatally diagnosed with VGAM have poor outcomes when cardiac or cerebral anomalies are present [12,13]. Conversely, those with isolated VGAM tend to have more favourable outcomes [14].

Management of VGAM should consist of aggressive intensive care management of cardiac failure and fluid status, as well as management of multi-organ dysfunction [2]. The treatment of choice for VGAM now consists of an endovascular approach to selectively occlude feeder vessels with coils or liquid embolic agents, aiming to reduce or eliminate the arteriovenous shunt [11,15]. This is typically achieved via the femoral transarterial route. Several procedures are often required to achieve a significant reduction in shunting [15]. Retrospective studies of patients treated with the endovascular approach report that between 56% and 85% of children are neurologically normal (follow-up period ranging from 6 months to 4 years) [2,14,15]. However, several of these studies included older infants and children in addition to neonates; these reported poorer outcomes in the neonatal group [14,15].

Lasjaunias and colleagues have developed a scale, the “Bicêtre neonatal evaluation score”, which evaluates the gross neurological status and non-neurological manifestations in neonates with VGAM and suggests an approach to treatment based on the patient’s score [14]. Using this system, our patient scored 8 out of 21, which falls into the emergency endovascular intervention category. Other considerations in the decision to treat our patient included the structurally normal appearance of the rest of the brain on MRI and near-normal bedside amplitude-integrated electroencephalogram (aEEG) monitoring.

## 4. Conclusions

This case highlights the difficulties in diagnosing a vein of Galen aneurysmal malformation in the context of another likely cause for persistent pulmonary hypertension and multi-organ dysfunction (in our case, perinatal asphyxia). It highlights the importance of considering high-output cardiac failure in neonates with persistent metabolic acidosis, cardiomegaly, and a structurally normal heart. Physical examination should include auscultation for a cranial bruit and a cranial ultrasound ordered early if there is clinical suspicion of high-output cardiac failure.

## 5. Learning Points

Vein of Galen aneurysmal malformations most commonly present in the neonatal period with high-output cardiac failure, often associated with persistent pulmonary hypertension and multi-organ dysfunction;A cranial bruit may be heard on clinical examination;Initial management should include aggressive intensive care management of fluid status, cardiac failure and any associated organ dysfunction, and referral to a tertiary paediatric centre;The treatment of choice for VGAM is endovascular occlusion of feeder vessels with coils or liquid embolic agents, and multiple such procedures are often required.

## Figures and Tables

**Figure 1 healthcare-12-00716-f001:**
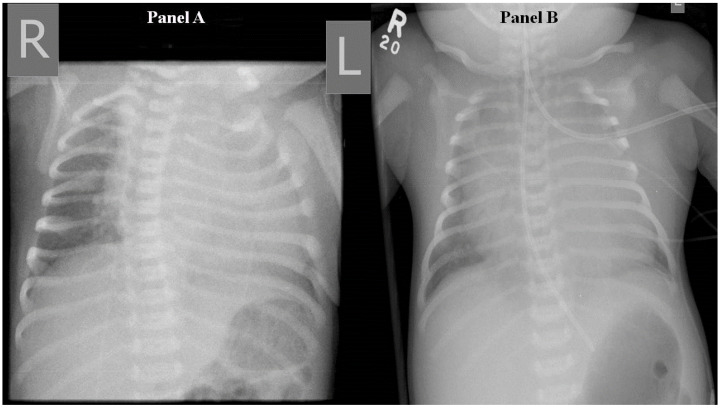
Frontal chest radiograph taken on day 1 demonstrating cardiomegaly predominantly in left chambers (Panel A) and worsening cardiomegaly on day 6 (Panel B). R denotes right side and L, left side.

**Figure 2 healthcare-12-00716-f002:**
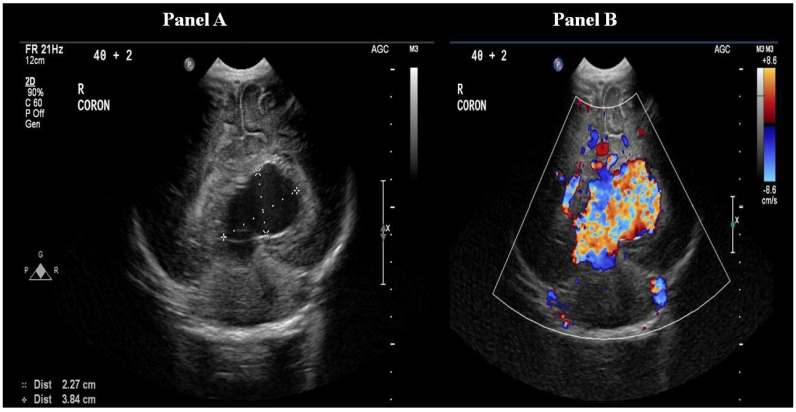
Coronal head ultrasound demonstrating an anechoic structure measuring 36 × 26 × 29 mm, consistent with dilated median prosencephalic vein (Panel A), and ultrasound with Doppler demonstrating increased flow in the dilated prosencephalic vein (Panel B).

**Figure 3 healthcare-12-00716-f003:**
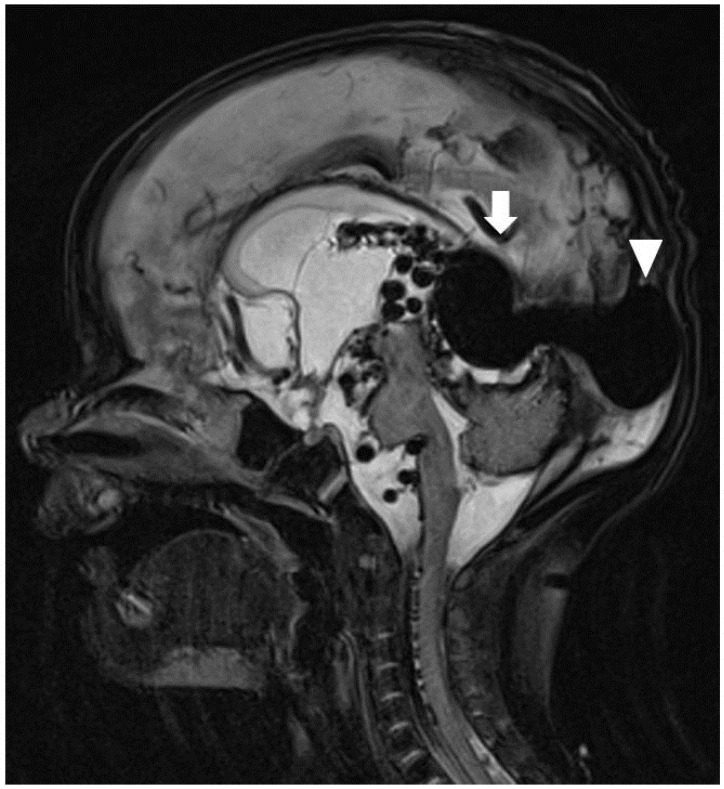
T2-weighted sagittal MRI. The dilated galenic vein, namely, the median vein of prosencephalon (arrow), located midline in the cistern of velum interpositum, drains into the superior sagittal sinus (arrowhead).

## Data Availability

The data presented in this study are available upon request from the corresponding author. The data are not publicly available due to privacy reasons.

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
