# Peer review of "Vein of Galen Aneurysmal Malformation: A Case Report"

_healthcare, 2024, doi:10.3390/healthcare12070716_

Round 1
Reviewer 1 Report
Comments and Suggestions for Authors
-the introduction is well organized as well as the embryology of VGAM.
-the case report is good but could use more information as to why this was confused with neonatal asphyxia, was it only blood gas?
-did the baby have hepatomegaly, other signs of heart failure, or an intracranial bruit (answered on line 91)
-the discussion is nice, you could highlight the mortality risk which is still high in this age group
-the conclusion of "look for AV malformations in shock with normal cardiac anatomy" (line 157-159) is well done
-line 35 remove "is"
Author Response
Response to Reviewer 1 Comments
Thank you very much for taking the time to review this manuscript. Please find the detailed responses below and the corresponding revisions/corrections highlighted/in track changes in the re-submitted files.
- Response to Comments on the Quality of English Language
Thank you for your feedback.
No specific action is needed.
- Response to Questions for General Evaluation
Thank you for your feedback.
No specific action is needed.
- Point-by-point response to Comments and Suggestions for Authors
Comments 1:
-the introduction is well organized as well as the embryology of VGAM.
Response: Thank you for your encouragement.
Comments 2:
-the case report is good but could use more information as to why this was confused with neonatal asphyxia, was it only blood gas?
Response:
Thank you for bringing this up. We modified the sentence slightly to read as, "The working diagnosis was perinatal asphyxia given the acidosis and multi-organ dysfunction". This will justify the thinking behind the working diagnosis.
Comments 3:
-did the baby have hepatomegaly, other signs of heart failure, or an intracranial bruit (answered on line 91)
Response: Thank you for pointing this out. Wee amended the sentence to read as, "Examination revealed worsening generalized oedema, ascites and hepatomegaly. This was attributed to congestive heart failure".
Comments 4:
-the discussion is nice, you could highlight the mortality risk which is still high in this age group
Response: Thank you for pointing this out. We added a couple of lines about mortality.
Comments 5:
-the conclusion of "look for AV malformations in shock with normal cardiac anatomy" (line 157-159) is well done
Response: Thank you for your confirmation.
Comments 6:
-line 35 remove "is"
Response: Thank you for pointing this out. We amended it as suggested.

Reviewer 2 Report
Comments and Suggestions for Authors
General comment
The authors described a neonate case who had VGAM, a rare cause of extracardiac shunt and was intensively treated including coil embolization. The reviewer almost agrees with the author’s explanation in the manuscript. However, there is a concern to resolve for the improvement of the manuscript.
Specific comment
Major comment 1
Although VGAM is a rare disease, the current case seems to undergo the common clinical scenario as the severe VGAM case. Can the author describe what is the novelty of the current case?
Minor comment 1
This case was born at home, but how was positive pressure ventilation applied immediately after birth?
Minor comment 2
Please describe the reason why this case was transferred to a tertiary care facility soon after birth.
Minor comment 3
Postnatal echocardiography was described as normal, but radiographs showed marked cardiac enlargement, and thus echocardiography would have also shown clear cardiac enlargement.
Minor comment 4
In lines 91 and 94 and Figure 2, the sizes of the dilated median prosencephalic vein are different (ex. 36x26x29, 26x28, 22.7x38.4), and thus they should be unified.
Author Response
Response to Reviewer 2 Comments
Thank you very much for taking the time to review this manuscript. Please find the detailed responses below and the corresponding revisions/corrections highlighted/in track changes in the re-submitted files.
- Response to Comments on the Quality of English Language
Thank you for your feedback.
No specific action is needed.
- Response to Questions for General Evaluation
Thank you for your feedback.
No specific action is needed.
- Point-by-point response to Comments and Suggestions for Authors
Comments 1:
The authors described a neonate case who had VGAM, a rare cause of extracardiac shunt and was intensively treated including coil embolization. The reviewer almost agrees with the author's explanation in the manuscript. However, there is a concern to resolve for the improvement of the manuscript.
Response: Thank you for your encouragement. We addressed the comments below.
Comments 2:
Although VGAM is a rare disease, the current case seems to undergo the common clinical scenario as the severe VGAM case. Can the author describe what is the novelty of the current case?
Response: We described this in the conclusion: "This case highlights the difficulties in diagnosing a Vein of Galen aneurysmal malformation in the context of another likely cause for persistent pulmonary hypertension and multi-organ dysfunction (in our case, perinatal asphyxia). It highlights the importance of considering high-output cardiac failure in neonates with persistent metabolic acidosis, cardiomegaly and a structurally normal heart. Physical examination should include auscultation for a cranial bruit and a cranial ultrasound ordered early if there is clinical suspicion of high-output cardiac failure".
Comments 3:
This case was born at home, but how was positive pressure ventilation applied immediately after birth?
Response: Thank you for pointing this out. Planned home delivery with midwifery support is not unusual in our region. It's available as a private option and on stricter criteria as a government-funded public programme.
To clarify this to the reader, we added: "A midwife attended the delivery at home".
Comments 4:
Please describe the reason why this case was transferred to a tertiary care facility soon after birth.
Response: We added the following to clarify this: "The patient was transferred to a tertiary paediatric hospital for paediatric neurosurgery and intervention radiology assessment as this service was unavailable at our hospital".
Comments 5:
Postnatal echocardiography was described as normal, but radiographs showed marked cardiac enlargement, and thus echocardiography would have also shown clear cardiac enlargement.
Response: The echo was done on the day of admission, i.e., early during the disease. It showed a 2.7mm patent ductus arteriosus (PDA) with bidirectional flow and suprasystemic pulmonary hypertension.
Comments 6:
In lines 91 and 94 and Figure 2, the sizes of the dilated median prosencephalic vein are different (ex. 36x26x29, 26x28, 22.7x38.4), and thus they should be unified.
Response: Thank you for pointing this out. We amended the measurement to be the same.

Reviewer 3 Report
Comments and Suggestions for Authors
This case exemplifies an intriguing presentation of Vein of Galen Aneurysmal Malformation. Despite the existence of documented instances of this pathology within the literature, including comprehensive case series, the authors are encouraged to expound upon the distinctive features of their particular case (i.e distinct or more intensive therapeutic interventions, differential diagnostic methodologies). Such elucidation is pivotal for enriching the existing comprehension of this condition within the scholarly discourse.
I will provide some articles which are already published in the literature:
https://www.ncbi.nlm.nih.gov/pmc/articles/PMC6473122/
https://www.ncbi.nlm.nih.gov/pmc/articles/PMC6295180/
Furthermore, it would be beneficial to furnish an illustration of the intraprocedural technique, and, if available, include echocardiographic images for enhanced visual representation.
Finally, I recommend expanding the discussion section by incorporating additional information pertaining to the pathology under consideration and its corresponding management strategies.
Comments on the Quality of English LanguageMinor English language revision.
Author Response
Response to Reviewer 3 Comments
Thank you very much for taking the time to review this manuscript. Please find the detailed responses below and the corresponding revisions/corrections highlighted/in track changes in the re-submitted files.
- Response to Comments on the Quality of English Language
Minor English language revision.
Thank you for your feedback.
We made more changes to clarify some of the sentences.
- Response to Questions for General Evaluation
Thank you for your feedback.
No specific action is needed.
- Point-by-point response to Comments and Suggestions for Authors
Comments 1:
This case exemplifies an intriguing presentation of Vein of Galen Aneurysmal Malformation. Despite the existence of documented instances of this pathology within the literature, including comprehensive case series, the authors are encouraged to expound upon the distinctive features of their particular case (i.e distinct or more intensive therapeutic interventions, differential diagnostic methodologies). Such elucidation is pivotal for enriching the existing comprehension of this condition within the scholarly discourse.
Response: Thank you for your suggestion. We agree with the reviewer that VGM is well-reported. However, we aimed to highlight the diagnostic challenge in such cases, particularly for junior doctors. We described this in the conclusion: "This case highlights the difficulties in diagnosing a Vein of Galen aneurysmal malformation in the context of another likely cause for persistent pulmonary hypertension and multi-organ dysfunction (in our case, perinatal asphyxia). It highlights the importance of considering high-output cardiac failure in neonates with persistent metabolic acidosis, cardiomegaly and a structurally normal heart. Physical examination should include auscultation for a cranial bruit and a cranial ultrasound ordered early if there is clinical suspicion of high-output cardiac failure".
Comments 2:
I will provide some articles which are already published in the literature:
https://www.ncbi.nlm.nih.gov/pmc/articles/PMC6473122/
https://www.ncbi.nlm.nih.gov/pmc/articles/PMC6295180/
Response: Thank you for pointing this out. We cited these two articles to strengthen our report.
Comments 3:
Furthermore, it would be beneficial to furnish an illustration of the intraprocedural technique, and, if available, include echocardiographic images for enhanced visual representation.
Response: Thank you for pointing this out. Unfortunately, we do not have illustrations for the intraprocedural technique at the tertiary centre. We agree that this would have strengthened the case it was planned earlier.
Comments 4:
Finally, I recommend expanding the discussion section by incorporating additional information pertaining to the pathology under consideration and its corresponding management strategies.
Response: Thank you for your comment. We added few lines about mortality. We documented the pathology (with references) in the introduction.

Round 2
Reviewer 2 Report
Comments and Suggestions for Authors
none
Author Response
There is no further comment from the reviewer. We addressed all the reviewer's comments satisfactorily.
We take this opportunity to thank the reviewer for their time, effort and expertise.
Reviewer 3 Report
Comments and Suggestions for Authors
The authors made all the necessary changes.
Author Response

(The authors gave the same response as above.)
